# Perceived Access to Health Care Services and Relevance of Telemedicine during the COVID-19 Pandemic in Germany

**DOI:** 10.3390/ijerph18147661

**Published:** 2021-07-19

**Authors:** Lukas Reitzle, Christian Schmidt, Francesca Färber, Lena Huebl, Lothar Heinz Wieler, Thomas Ziese, Christin Heidemann

**Affiliations:** 1Department of Epidemiology and Health Monitoring, Robert Koch Institute, 12101 Berlin, Germany; SchmidtChri@rki.de (C.S.); FaerberF@rki.de (F.F.); ZieseT@rki.de (T.Z.); HeidemannC@rki.de (C.H.); 2Department of Tropical Medicine, Bernhard Nocht Institute for Tropical Medicine, University Medical Center Hamburg-Eppendorf, 20359 Hamburg, Germany; l.huebl@uke.de; 3I. Department of Medicine, University Medical Center Hamburg-Eppendorf, 20251 Hamburg, Germany; 4Robert Koch Institute, 13353 Berlin, Germany; WielerLH@rki.de

**Keywords:** COVID-19, SARS-CoV-2, access to health care, health care utilization, telemedicine

## Abstract

During the COVID-19 pandemic in Germany, non-pharmaceutical interventions were imposed to contain the spread of the virus. Based on cross-sectional waves in March, July and December 2020 of the COVID-19 Snapshot Monitoring (COSMO), the present study investigated the impact of the introduced measures on the perceived access to health care. Additionally, for the wave in December, treatment occasion as well as utilization and satisfaction regarding telemedicine were analysed. For 18–74-year-old participants requiring medical care, descriptive and logistic regression analyses were performed. During the less strict second lockdown in December, participants reported more frequently ensured access to health care (91.2%) compared to the first lockdown in March (86.8%), but less frequently compared to July (94.2%) during a period with only mild restrictions. In December, main treatment occasions of required medical appointments were check-up visits at the general practitioner (55.2%) and dentist (36.2%), followed by acute treatments at the general practitioner (25.6%) and dentist (19.0%), treatments at the physio-, ergo- or speech therapist (13.1%), psychotherapist (11.9%), and scheduled hospital admissions or surgeries (10.0%). Of the participants, 20.0% indicated utilization of telemedical (15.4% telephone, 7.6% video) consultations. Of them, 43.7% were satisfied with the service. In conclusion, for the majority of participants, access to medical care was ensured during the COVID-19 pandemic; however, access slightly decreased during phases of lockdown. Telemedicine complemented the access to medical appointments.

## 1. Introduction

The COVID-19 pandemic affects the health care sector in many ways. On the one hand, health care providers and health care systems are facing new challenges directly through the care of patients with COVID-19. Treatment of a new disease needs to be established while evidence-based recommendations on medical management are limited and constantly updated [1,2]; at the same time, scarce capacities of staff, intensive care beds and medical equipment have to be dealt with at the point of rising numbers of COVID-19 patients [1]. On the other hand, extensive non-pharmaceutical interventions (NPI) have been introduced to contain the spread of severe acute respiratory syndrome corona virus type 2 (SARS-CoV-2). In many countries, non-COVID-19 treatments have been restricted to ensure capacities for treatment of severe cases of COVID-19 [3,4,5]. As a result, also in Germany, elective surgical interventions have been postponed and cancelled [6,7,8], services for routine care such as health check-ups, screenings for cancer or vaccinations have decreased [9,10,11,12,13,14,15,16]. Furthermore, a decline in primary care contacts at general practitioners (GP) or specialists have been reported [17,18]. Studies have indicated that the utilization of emergency care for acute conditions like myocardial infarction or ischemic stroke decreased during the first months of the COVID-19 pandemic in Germany [8,19,20,21] and other countries [22,23,24].

The effects of NPI on medical care have also been investigated from the population’s point of view. On the one hand the analyses of previous waves of the COVID-19 Snapshot Monitoring (COSMO) study showed that the majority of participants reported no difficulties regarding access to medical appointments; however, this proportion was reduced during the first lockdown compared to July 2020, when measures have been loosened [9,25]. On the other hand findings suggest that a reduced demand for patients’ care may have additionally led to decreased utilization of health care services [26] as some people postponed and avoided routine care and even urgent care because of concerns of COVID-19 [27]. Especially insecurities and people’s fear of exposure to SARS-CoV-2 could play a role in this as well as the intention to not overburden the health care system [7,28,29]. In consequence, negative effects of the pandemic on health status are also feared for persons without COVID-19 disease, particularly for those with chronic medical conditions [30,31]. Furthermore, the extent to which the provision of telemedical services is being expanded to deal with the challenges of health care in the pandemic is an important topic for further research. Even though telemedicine has limited applicability, it could to some extent mitigate the effects of postponed or cancelled doctor’s appointments and in many countries a rise in utilization of telemedicine has been reported [32].

In summary, there have been a number of studies on access to health care services in Germany during the first months of the COVID-19 pandemic; however, findings on the availability of medical care throughout the further course of the pandemic are yet scarce, but strongly relevant [33]. In Germany, in mid-March 2020, a set of restrictions to reduce physical contact was implemented, including the ban of gatherings of more than two people, a minimum distance of 1.5 metres between people in public, the closing of schools and day care centres as well as the closure of shops and restaurants [34]. Concurrently, hospitals were asked to postpone scheduled surgeries and treatments to provide capacity for treatment of severe cases of COVID-19 [6]. From the end of April, measures were loosened gradually so that by May restaurants and schools in many German federal states could be reopened [34]. After a period of mild restrictions during summer, re-implementation of restrictions started from the 2nd of November in several stages. In December, contact restrictions were re-enforced; however, schools and shops were not yet affected (“less strict second lockdown”) [34].

Against this background, one main objective of our study was to examine perceived access to medical care during three different stages of lockdown over the course of the COVID-19 pandemic in 2020, i.e., in March after the first implementation of measures to contain the spread of SARS-CoV-2 (“first lockdown”), in July after the loosening of restrictions (“period of mild restrictions”) and at the beginning of December in a phase of reintroduction of regulations though less strict than in spring (“less strict second lockdown”). Additionally, we present stratified results on the required treatment occasion as well as the use and satisfaction regarding telemedicine services such as phone or video calls instead of physical visits for the stage of the second lockdown from a population-based perspective.

## 2. Materials and Methods

### 2.1. Study Sample

The present study is based on data from the serial, cross-sectional COVID-19 Snapshot Monitoring (COSMO) online survey, initiated in Germany in March 2020. For rapid and adaptive monitoring of knowledge, risk perceptions, preventive behaviours, and public trust in the context of the COVID-19 pandemic across countries, the initiating authorities together with the new Insights Unit at the WHO Regional Office for Europe are providing a customizable study protocol, a sample questionnaire, a script for data analysis as well as directions on contextual adaptation and open access procedures [35,36].

In Germany, the COSMO survey was conducted weekly between 3/4March 2020 and 26/27 May 2020 (waves 1 to 13) and bi-weekly or weekly thereafter (wave 14 to current wave 43) [37]. For this analysis, data from wave 6 (7/8 April 2020, *n* = 1022), wave 17 (21/22 July 2020, *n* = 1001), and wave 28 (1/2 December 2020; *n* = 1020) were included. Participants are German-speaking, living in Germany and aged 18 to 74 years. The sample for each wave is matching the general population in Germany in terms of age, gender, and residency in a German federal state (Nielsen areas) [35,36]. Participants for whom the initial question about currently required medical appointments was not applicable were excluded, resulting in a final sample size of 773 participants for wave 6941 participants for wave 17, and 868 participants for wave 28.

### 2.2. Assessment of Variables Related to Access to Health Care Services and Telemedicne

In waves 6, 17, and 28, participants were asked: “Are doctor visits and contacts necessary for you currently available?” (response options: yes; no; does not apply).

In wave 28, participants were further asked: (1) “What is the reason for doctor visits or contacts that are currently necessary for you?” (response options, whereby multiple answers were possible: psychotherapeutic treatment; physiotherapeutic, ergotherapeutic or speech therapy treatment; check-up visit at general practitioner or specialist; check-up visit at dentist; acute (urgent) treatment at general practitioner or specialist; acute (urgent) treatment at dentist; scheduled hospital admission or surgery; other [free text entry possible]) and (2) “Do you currently use telephone or telemedical contact options instead of visiting a doctor’s or psychotherapist’s practice?” (response options, whereby multiple answers were possible: yes, telephone, e.g., telephone consultations; yes, telemedicine, e.g., video consultations, e-mail contact; no; no need for examination or treatment). If the latter question was answered with yes regarding the telephone or telemedical contact, participants were further asked: (3) “Are you satisfied with your telephone or telemedical consultation?” (response options: completely unsatisfied to completely satisfied on a seven-point scale).

### 2.3. Assessment of Sociodemographic Variables and Presence of Chronic Diseases

In accordance to previous studies based on COSMO data, age was categorized into the following four categories: 18 to 29 years, 30 to 49 years, 50 to 64 years, and 65 to 74 years, but education was maintained in the assessed three categories: up to 9 years of education, at least 10 years of education (without general qualification for university entrance), and at least 10 years of education (with general qualification for university entrance) [9,25]. The presence of chronic conditions was assessed by asking: “Do you have a chronic disease?” (response options: yes; no; don’t know).

### 2.4. Statistical Analysis

Descriptive analyses include the calculation of frequencies and proportions (95% confidence interval) of sample characteristics and health care services variables. The logistic regression model considers the perceived access to medical appointments as a binary dependent variable and the sociodemographic variables as well as the presence of a chronic condition as independent variables scaled as factors. The criterion for statistical significance was set at *p* < 0.05. SAS 9.4 (SAS Institute, Cary, NC, USA) was used for all statistical analyses.

### 2.5. Ethics Statement

Informed consent was obtained from all participants included in the study. All procedures conducted in the COSMO study are in accordance with the ethical standards of the University of Erfurt institutional research committee [35] and with the 1964 Helsinki Declaration and its later amendments or comparable ethical standards [25].

## 3. Results

### 3.1. Sample Characteristics

Across the three waves in April, July and December 2020, the participants requiring access to medical appointments were comparable with regard to sex, age, education and self-reported presence of a chronic condition (Table 1). The distribution of age and sex (crossed) in the sample is in accordance with data of the German population provided by the last German census conducted in 2011 [38] (Appendix A). Regarding education, more than half of the participants reported to have more or equal to 10 years of education with a university entrance qualification. Chronic conditions were reported by almost 40% of the individuals.

### 3.2. Perceived Access to Medical Appointments during the COVID-19 Pandemic

In December 2020, the majority of participants indicated to have access to medical appointments (Table 2). Regarding age groups, the proportion of participants reporting ensured access to medical appointments increased significantly with older age. While 84.4% of those aged 18–29 reported ensured access, 94.4% and 97.8% in the 50–64 and 65–74 age groups, respectively, agreed with this statement. Confidence intervals did not overlap between respective groups. Sex, educational background and presence of a chronic condition had no significant influence on the perceived access to medical appointments.

During the less strict second lockdown from the end of October until the beginning of December 2020 in Germany, the perceived access to medical appointments was higher than during the first stricter lockdown in April 2020 and lower than in July 2020 during the period with only mild restrictions (Table 2). Consistent with wave 28 in December 2020, both previous waves show the same patterns for the stratified analysis. While there were no differences observed regarding sex, education and the presence of chronic conditions, increasing age of the participants was associated with higher reported access to medical appointments.

To assess whether the availability of medical services differs between the three phases of the pandemic independent of effects described above, we performed a logistic regression model adjusting for multiple covariates sex, age, education and presence of a chronic condition. The logistic regression model confirmed the observed differences (Appendix A). Compared to wave 28 in December 2020, the perceived access to medical appointments was higher in July 2020 (OR: 1.57, 95%-CI: 1.09–2.26) and lower in April 2020 (OR: 0.62, 95%-CI: 0.45–0.85). The odds to have ensured access to health care were not decreased for participants with self-reported chronic conditions compared to participants without chronic conditions (Appendix A). Regarding sex and education, no significant differences were observed. However, with increasing age, the odds of reporting ensured access to medical appointments increased. Defining the age group 18–29 years as reference, the age groups 30–49 years (OR: 1.87, 95%-CI: 1.34–2.62), 50–64 years (OR: 3.39, 95%-CI: 2.25–5.13) and 65–74 years (OR: 6.65, 95%-CI: 3.66–12.08) showed continuously increasing odds ratios for ensured access to medical appointments.

### 3.3. Treatment Occasions

Within wave 28 in December 2020, we additionally assessed the occasion for a required medical appointment (Table 3). More than half of the participants reported that check-up visits at their general practitioner (GP) or their specialist were required. More than one third indicated that a check-up visit at their dentist was required. An acute (urgent) treatment at GP or specialist was relevant for a quarter of participants and an acute treatment at dentist was relevant for almost a fifth of participants. Further, treatments at a physiotherapist, ergotherapist or speech therapist or treatments at a psychotherapist were less frequently observed. One in ten individuals stated that a scheduled hospital admission or surgery was required. Other treatment occasions were indicated by 9.0% of respondents and according to the sighted free text entries mainly comprised appointments to receive prescriptions, vaccinations or to participate at screening examinations.

The stratified analysis revealed that check-up visits at the GP or specialist were more frequently required by participants of older age and with chronic conditions (Table 3). Interestingly, for acute treatment occasions at the GP or specialist there were no differences observed between age groups or the presence of a chronic condition.

Regarding check-up visits at the dentist, individuals in younger age groups and with higher educational background reported more frequently that appointments were necessary compared to older age groups and participants with lower educational background. In contrast, for acute treatment occasions at the dentist no differences with respect to age and educational background were observed.

For psychotherapy, participants with chronic conditions more often indicated that treatments were required. Similarly, participants with chronic conditions more frequently reported that physiotherapy, ergotherapy or speech therapy was required. Men and individuals of younger age groups required treatments in the hospital more often than women or older age groups, respectively. With respect to all other strata, no differences were observed for occasions of required medical appointment.

As described above, during the second lockdown in December 2020 for the majority of participants access to medical appointments was ensured. The stratified analysis confirms this observation for all treatment occasions (Figure 1). For example, out of all participants requiring check-up visits at the GP or specialist (55.2%) or dentist (36.3%) (Table 3), the medical appointment was available for 51.7% or 34.0%, respectively. Contrarily, 3.5% and 2.3% of respondents, respectively, reported they had no access to the required medical appointment. For all other treatment occasions, fewer participants indicated the requirement as well as no access to medical appointments.

However, stratified analysis considering the relative proportion of individuals without access to medical appointment for each required treatment occasion separately revealed a different picture. More specifically, 16.5% of respondents requiring psychotherapy had no access to an appointment. For scheduled hospitals admissions or operations, 11.5% of individuals and for physiotherapy, ergotherapy or speech therapy, 9.6% of individuals reported their respective appointment was not available. Of participants requiring an acute appointment at their GP or specialist or at their dentist, this proportion was 8.2% or 8.4%, respectively; of participants requiring a check-up visit at their GP or specialist or at their dentist, this proportion was lowest with 6.3% each.

### 3.4. Utilization of Telemedcine during the COVID-19 Pandemic

In wave 28 in December 2020, the utilization of telemedicine for the interaction of physicians with patients via audio and video during the pandemic was additionally assessed. Overall, 20.0% of the participants requiring a medical appointment indicated that they had a consultation via telephone, video or both (Appendix A). The remaining participants either did not utilize telemedical services or had no need. Appointments via telephone (15.4%) were more frequently reported than video (7.6%). Men utilized both telephone and telemedicine more frequently than women. While there was no difference observed regarding the telephone appointments across age groups and educational groups, video consultations were more frequently utilized by participants of younger age and with a higher educational background. With regard to chronic condition no differences in the utilization of telemedical consultations were reported. Respondents who indicated no access to medical appointments used telemedicine more frequently. Within the group of participants who had no need for telemedical consultations despite requiring a medical appointment there were no differences between sex, age and educational groups observed.

When asked about their satisfaction with telemedicine consultations, overall 43.7% were satisfied (6 or 7 points out of 7 points), 48.9% were neither fully satisfied nor dissatisfied (3 to 5 points) and 7.5% were not satisfied (1 to 2 points) (Figure 2). There were no relevant differences in satisfaction between telephone and video consultations. Women and men were equally content with the telemedicine appointments. Individuals between 18–29 years were less frequently fully satisfied (23.8%) with the telemedicine appointments compared to the other age groups. Participants aged 65–74 years and those with chronic conditions more often reported that they were dissatisfied with the telemedicine appointments.

## 4. Discussion

The present study based on data of the population-based COSMO survey shows that during the course of the COVID-19 pandemic in 2020, access to medical care services was available for the majority of 18- to 74-year-olds reporting a need for medical care. However, the proportion of those who reported non-availability of needed medical care services was corresponding to the strictness of NPI that have been introduced for the containment of SARS-CoV-2. While check-up visits and acute treatments at the general practitioner or specialist and at the dentist were the most frequently required treatments, psychotherapy sessions were the most frequently non-available treatment when considered in relation to the respective frequency of required treatments. Remarkably, one-fifth of participants utilized telephone or video consultation instead of visiting a physician’s or psychotherapist’s practice, and this proportion was higher among those who reported non-availability of medical care services. Some differences were observed by age, sex, educational background or the presence of a chronic condition.

### 4.1. Perceived Access to Medical Appointments during the COVID-19 Pandemic

According to an earlier analysis of the COSMO study, the vast majority stated to have access to required medical appointments during the first months of the COVID-19 pandemic, although this proportion was lower in April 2020 (86.8%), during the first lockdown, compared to July 2020 (94.2%), when most NPIs were lifted [9]. This is in line with another study based on COSMO data from April and July 2020 reporting that participants mostly denied having difficulties to access medical care [25]. Still, results from a recent nationwide German Health Update (GEDA 2019/2020) survey covering data from the beginning of the COVID-19 pandemic until September 2020 indicate a temporary decline in both outpatient visits at general practitioners (GP) and specialists between April and June 2020 compared to the same months in 2019 [18]. The present study adds that at the beginning of the less strict second lockdown in December 2020, the perceived access to medical appointments was again reduced (91.2%); however, not to the same extent as during the first lockdown. NPIs such as mask-wearing, enhanced hygiene, contact reductions, ban of large public events, recommendations to work from home or for testing and isolation were in place throughout the pandemic, while closing educational institutions, restaurants, shopping facilities, border controls, and travel restrictions varied across time and federal state level [34]. An important indicator of lockdown measures and their acceptance is overall mobility [39]. Studies showed that in the months when most NPIs were lifted or gradually loosened (May to October), mobility in Germany increased to a level comparable to the previous year [40]. With the start of the second lockdown, mobility fell back below the level of the previous year [41].

Younger participants indicated more often to have no access to medical appointments throughout all three points in time. Previously, a study on access to health care in Europe reported younger people having challenges in access to medical appointments [42] and a population-based survey from the Netherlands and Belgium carried out during the first lockdown confirmed this observation [43]. Although several studies further report that scheduled treatments for patients with chronic diseases were frequently cancelled by health care providers [30,31], in the present study, participants with reported chronic conditions did not report a lower perceived access to health care. Interestingly, results from the nationwide GEDA 2019/2020 survey in Germany showed that in spring of 2020 compared to spring of 2019, there was a temporary decrease in specialist utilization, but not in GP utilization among people with diabetes [44], whereas in the overall population, utilization was temporarily decreased in both specialist and GP utilization [18]. This was discussed as possibly being related to adaptions in the health care services by telephone consultations, which were also captured by the study question on GP utilization, for people with regularly required care of their GP [44].

### 4.2. Treatment Occasions

With regard to the relevant treatment occasions assessed in December 2020, the present study revealed that the most required health care services were check-up visits at the GP, specialist or dentist, which were also most frequently reported not to be available. This is in line with previous reports highlighting that especially regular screening visits for cancer and preventive medical check-ups were impacted by the lockdown in Germany [9,10]. This has also been reported in other countries like the UK, Australia and Taiwan [11,12,13]. Further research will determine whether postponed check-up and screening visits will have a substantial impact on long term health outcomes like delayed diagnosis as reported in several countries [28,45,46] or poorer health outcomes for already diagnosed chronic diseases [47]. Scheduled hospital admissions or surgeries were less frequently required by participants; however, more than one out of ten planned appointments were not available. This is in line with an analysis of claims data on the hospital admissions during the COVID-19 pandemic in Germany [21]. While during the first lockdown in March and April 2020, a strong decrease up to one third of admissions compared to the same time period in 2019 has been observed, in November and December 2020, the decrease in utilization was less severe, ranging from 17 to 20%.

However, not only scheduled check-up visits or hospital admissions, but also urgent treatments were required and, according to the participants, not always available. An extensive analysis from the UK showed that primary care contacts for almost all acute physical and mental health conditions were significantly reduced during the lockdown in March 2020 [17]. The most profound impact was observed for consultations for mental health conditions. The present study also revealed an impact of the lockdown on the access to psychotherapy treatments, for which the proportion of individuals reporting non-availability of appointments was higher (16.5%) than for other surveyed treatment occasions (between 6.3% and 11.5%). As in Germany already prior to the pandemic, waiting times for psychotherapy were higher compared to other specialties [48], the direct effect of the lockdown on the availability of psychotherapy appointments cannot be differentiated. However, the lockdown not only challenged the continuation of care of patients with previously diagnosed psychiatric conditions, but also increased the need for consultation for mental health complaints as a consequence of the lockdown and the COVID-19 pandemic itself [49]. Several systematic reviews report that among others, the frequency of symptoms of anxiety, depression, insomnia and psychological distress have significantly increased in the general population during the COVID-19 pandemic [50,51,52]. In Germany, the impact of the pandemic on mental health is less clear. An early analysis of the German National Cohort showed an increase in depressive symptoms, especially in young people [53], which, however, was not seen in the GEDA study [18]. To date, no clear evidence for an increase in self-harming and suicidal behaviour has been observed [54]. In the early stage of the pandemic, even a decrease in suicide rates was reported; however, this was followed by an increase later in the pandemic. Therefore, the access to health care for prevention and treatment of mental health problems should be one priority, and continuous research on the topic is required [55,56].

### 4.3. Utilization and Satisfaction Regarding Telemedcine during the COVID-19 Pandemic

Telemedicine offers a broad range of opportunities to face the challenge in delivery of care during the COVID-19 pandemic and can especially facilitate the interaction between physicians and patients [32,57]. On the one hand, telemedicine enables contactless consultation with patients having respiratory symptoms and reduces the risk for health care professionals. On the other hand, telemedicine serves to reduce the risk of acquiring an infection for appointments not related to COVID, especially for vulnerable groups such as the chronically ill [58]. Additionally, in Germany, utilization of video consultation rose significantly [10]. While in 2019, according to claims data from 70 million persons covered by statutory health insurance only 200 video consultations per month were provided, in 2020, more than 200,000 video consultations were documented. For psychotherapy treatments, a survey from three European countries including Germany also showed an increase in the utilization of video and telephone consultations [59]. In the present study, we analysed the use of these telemedical consultation possibilities also stratified by personal characteristics and further asked for the satisfaction with this alternative contact option. Our results confirm that people with a required access to health care services reported a relatively high utilization of telephone or video consultations, especially if perceived access was limited. Only 7.5% of participants were not satisfied with these consultations. During the COVID-19 pandemic, telemedicine was useful in many different medical disciplines and both patients and healthcare providers reported high satisfaction [60].

However, our data also shows that access to telemedicine was not equally distributed. Men stated to use both telephone and video consultations more often than women, and participants with a higher educational background and of younger age reported to use video consultations instead of physical visits more often than the other respective participant subgroups. The latter findings are in line with a systematic review on the access to telemedicine, which identified suitable technical equipment and familiarity with the software systems as major facilitators of the telemedicine [61]. Telemedicine, however, can only complement doctor visits in primary care, especially if examination, imaging or lab tests are necessary either for the assessment of acute symptoms and screening or long-term care. Yet, the increasing access to non-physical contacts via telephone and video during the pandemic could push digitalization in the health service sector [32]. Video consultation might become an integral part of doctor-patient communication in the future and ensure the quality of care, especially in areas with less developed care structures; however, equal access independent of age and educational status and high quality of care must be guaranteed.

### 4.4. Strength and Limitations

The COSMO study provides timely data on the perception, attitudes, knowledge and behaviour of the general population in Germany during the COVID-19 pandemic. The study is designed as a representative sample of the population aged 18 to 74 years. However, a selection bias due to the relatively low response (<20%) [62] and the online mode of the study cannot be excluded and might influence the generalizability of the study results for the adult German population. Due to stratification the sample size in certain strata was below a recommended threshold needed for reliable estimation of confidence intervals [63]. Instead of implementing methods for calculating confidence intervals for rare events [63] or sum up groups for higher statistically robustness, we preferred being cautious with our interpretations. That is, that we used significantly in a statistical meaning only for comparisons of strata with more than 30 persons. With regard to the online sampling, persons reached might be more inclined to the utilization of telemedicine services. Additionally, the elderly population is not included in the study; however, these participants might be especially vulnerable in their need of access to health care and are less likely to use telemedical services. Furthermore, similar to other population-based surveys, multimorbid or institutionalized persons are hard to reach and therefore likely to be underrepresented in the study.

## 5. Conclusions

The results of the COSMO study indicate that for the majority of the German population aged 18–74 years medical appointments were available throughout the COVID-19 pandemic. However, perceived availability was reduced during lockdown periods and corresponded to the extent of the restrictions. Although, most persons with chronic conditions have not reported limited access to health care, further analyses are required to assess the impact of the lockdown for specific disease groups. Especially, the reduced availability and utilization of preventive medical check-up visits, screenings and psychotherapy treatments could have an effect on long term health outcomes such as the early diagnosis of cancer and the quality of care of people with chronic diseases. The increased use of telemedicine can complement the in-person visits and the COVID-19 pandemic could accelerate its implementation, which should focus on equal access to telemedical offers regardless of age or education.

## Figures and Tables

**Figure 1 ijerph-18-07661-f001:**
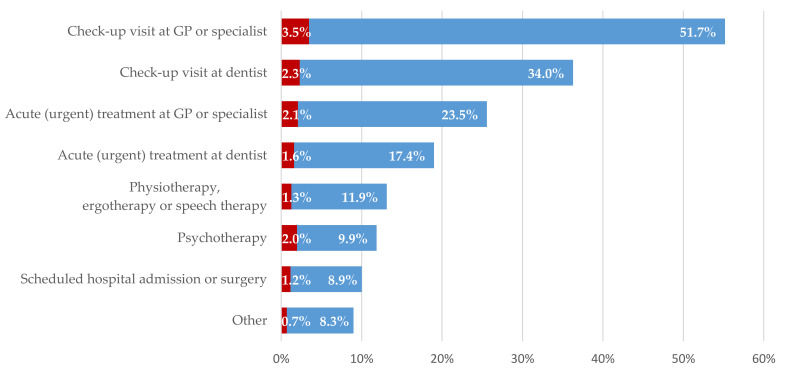
Proportion of treatment occasions among participants requiring medical appointments stratified by perceived non-availability (red) and availability (blue) of appointments (wave 28, *n* = 868). GP: general practitioner.

**Figure 2 ijerph-18-07661-f002:**
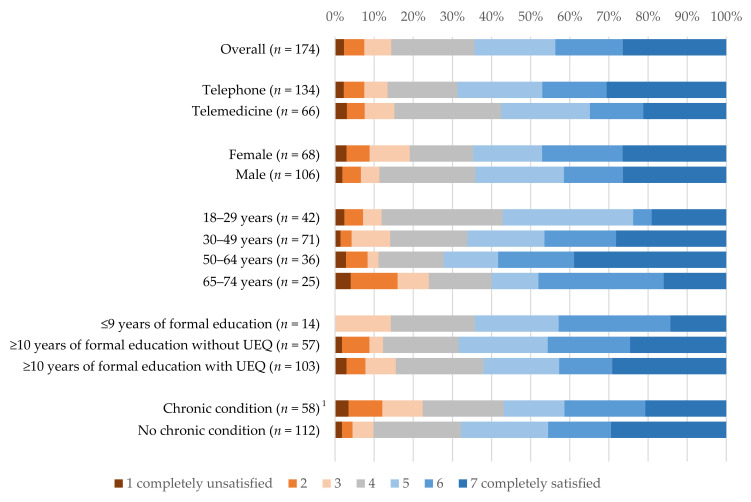
Satisfaction of participants having used telemedicine (wave 28, *n* = 174) ^1^
*n* = 4 participants indicated “don’t know”. UEQ: university entrance qualification.

**Table 1 ijerph-18-07661-t001:** Characteristics of individuals requiring access to medical appointments in waves 6, 17 and 28.

	Wave 6 April 2020 (*n* = 773)		Wave 17 July 2020 (*n* = 941)		Wave 28 December 2020 (*n* = 868)	
	% (95%-CI)	*n*	% (95%-CI)	*n*	% (95%-CI)	*n*
Sex						
Male	50.2 (46.7–53.7)	388	48.4 (45.2–51.6)	455	47.7 (44.4–51.0)	414
Female	49.8 (46.3–53.3)	385	51.6 (48.4–54.8)	486	52.3 (49.0–55.6)	454
Age groups						
18–29 years	16.4 (14.0–19.2)	127	18.8 (16.4–21.4)	177	17.7 (15.3–20,4)	154
30–49 years	38.2 (34.8–41.6)	295	35.7 (32.7–38.8)	336	37.6 (34.4–40.8)	326
50–64 years	28.3 (25.3–31.6)	219	28.3 (25.5–31.2)	266	28.8 (25.9–31.9)	250
65–74 years	17.1 (14.6–19.9)	132	17.2 (14.9–19.8)	162	15.9 (13.6–18.5)	138
Education						
≤9 years of formal education	9.8 (7.9–12.1)	76	13.5 (11.5–15.8)	127	12.1 (10.1–14.4)	105
≥10 years of formal education without university entrance qualification	35.7 (32.4–39.2)	276	32.6 (29.7–35.7)	307	33.4 (30.3–36.6)	290
≥10 years of formal education with university entrance qualification	54.5 (50.9–57.9)	421	53.9 (50.7–57.0)	507	54.5 (51.2–57.8)	473
Chronic condition						
Yes	39.7 (36.3–43.2)	307	38.4 (35.3–41.5)	361	37.0 (33.8–40.2)	321
No	55.5 (52.0–59.0)	429	59.6 (56.4–62.7)	561	59.8 (56.5–63.0)	519
Don’t know	4.8 (3.5–6.5)	37	2.0 (1.3–3.1)	19	3.2 (2.2–4.6)	28

95%-CI: 95% confidence interval.

**Table 2 ijerph-18-07661-t002:** Proportion of participants reporting ensured access to medical appointments among participants requiring medical appointments in waves 6, 17 and 28 stratified by sex, age, education and chronic condition.

	Wave 6 April 2020 (*n* = 773)		Wave 17 July 2020 (*n* = 941)		Wave 28 December 2020 (*n* = 868)	
	% (95%-CI)	*n*	% (95%-CI)	*n*	% (95%-CI)	*n*
Overall	86.8 (84.2–89.0)	671	94.2 (92.5–95.5)	886	91.2 (89.2–93.0)	792
Sex						
Male	86.1 (82.3–89.2)	334	93.6 (91.0–95.5)	426	90.6 (87.4–93.0)	375
Female	87.5 (83.8–90.5)	337	94.7 (92.3–96.3)	460	91.9 (89.0–94.0)	417
Age groups						
18–29 years	79.5 (71.6–85.7)	101	87.0 (81.2–91.2)	154	84.4 (77.8–89.3)	130
30–49 years	86.8 (82.4–90.2)	256	92.9 (89.6–95.2)	312	89.3 (85.4–92.2)	291
50–64 years	87.7 (82.6–91.4)	192	97.7 (95.1–99.0)	260	94.4 (90.8–96.7)	236
65–74 years	92.4 (86.5–95.9)	122	98.8 (95.2–99.7)	160	97.8 (93.5–99.3)	135
Education						
≤9 years of formal education	89.5 (80.3–94.7)	68	94.5 (88.9–97.4)	120	90.5 (83.2–94.8)	95
≥10 years of formal education without university entrance qualification	87.0 (82.4–90.4)	240	93.8 (90.5–96.0)	288	90.7 (86.8–93.5)	263
≥10 years of formal education with university entrance qualification	86.2 (82.6–89.2)	363	94.3 (91.9–96.0)	478	91.8 (88.9–93.9)	434
Chronic condition						
Yes	87.6 (83.4–90.9)	269	94.2 (91.2–96.2)	340	91.3 (87.7–93.9)	293
No	86.0 (82.4–89.0)	369	94.1 (91.8–95.8)	528	91.5 (88.8–93.6)	475
Don’t know	89.2 (74.5–95.9)	33	94.7 (70.6–99.3)	18	85.7 (67.5–94.5)	24

95%-CI: 95% confidence interval.

**Table 3 ijerph-18-07661-t003:** Proportion of treatment occasions among participants requiring medical appointments stratified by sex, age, education and chronic condition (wave 28, n = 868).

	Psychotherapy		Physiotherapy, Ergotherapy or Speech Therapy		Check-Up Visit at GP or Specialist		Check-Up Visit at Dentist		Acute (Urgent) Treatment at GP or Specialist		Acute (Urgent) Treatment at Dentist		Scheduled Hospital Admission or Surgery	
	% (95%-CI)	*n*	% (95%-CI)	*n*	% (95%-CI)	*n*	% (95%-CI)	*n*	% (95%-CI)	*n*	% (95%-CI)	*n*	% (95%-CI)	*n*
Overall	11.9 (9.9–14.2)	103	13.1 (11.0–15.6)	114	55.2 (51.9–58.5)	479	36.3 (33.1–39.6)	315	25.6 (22.8–28.6)	222	19.0 (16.5–21.8)	165	10.0 (8.2–12.2)	87
Sex														
Male	11.8 (9.1–15.3)	49	11.6 (8.8–15.1)	48	52.4 (47.6–57.2)	217	36.7 (32.2–41.5)	152	26.1 (22.1–30.5)	108	21.7 (18.0–26.0)	90	12.6 (9.7–16.1)	52
Female	11.9 (9.2–15.2)	54	14.5 (11.6–18.1)	66	57.7 (53.1–62.2)	262	35.9 (31.6–40.4)	163	25.1 (21.3–29.3)	114	16.5 (13.4–20.2)	75	7.7 (5.6–10.6)	35
Age groups														
18–29 years	13.6 (9.1–20.0)	21	8.4 (5.0–14.0)	13	55.2 (47.3–62.9)	85	41.6 (34.0–49.5)	64	24.7 (18.5–32.1)	38	20.8 (15.1–27.9)	32	14.9 (10.1–21.5)	23
30–49 years	13.8 (10.5–18.0)	45	16 (12.4–20.3)	52	46.9 (41.6–52.4)	153	44.5 (39.2–49.9)	145	27.3 (22.7–32.4)	89	19.3 (15.4–24.0)	63	12.0 (8.9–16.0)	39
50–64 years	11.2 (7.8–15.8)	28	14 (10.2–18.9)	35	58.0 (51.8–64.0)	145	27.6 (22.4–33.5)	69	24.4 (19.5–30.1)	61	21.2 (16.6–26.7)	53	8.0 (5.2–12.1)	20
65–74 years	6.5 (3.4–12.1)	9	10.1 (6.1–16.4)	14	69.6 (61.4–76.7)	96	26.8 (20.1–34.8)	37	24.6 (18.2–32.5)	34	12.3 (7.8–18.9)	17	3.6 (1.5–8.4)	5
Education														
≤9 years of formal education	10.5 (5.9–18.0)	11	8.6 (4.5–15.7)	9	59.0 (49.4–68.0)	62	26.7 (19.1–35.9)	28	27.6 (19.9–36.9)	29	21.9 (15.0–30.8)	23	15.2 (9.5–23.5)	16
≥10 years of formal education without university entrance qualification	10.0 (7.0–14.0)	29	12.1 (8.8–16.4)	35	55.2 (49.4–60.8)	160	28.6 (23.7–34.1)	83	25.9 (21.1–31.2)	75	16.9 (13.0–21.7)	49	9.0 (6.2–12.9)	26
≥10 years of formal education with university entrance qualification	13.3 (10.5–16.7)	63	14.8 (11.9–18.3)	70	54.3 (49.8–58.8)	257	43.1 (38.7–47.6)	204	24.9 (21.2–29.1)	118	19.7 (16.3–23.5)	93	9.5 (7.2–12.5)	45
Chronic condition														
Yes	15.9 (12.3–20.3)	51	19.3 (15.4–24.0)	62	64.5 (59.1–69.5)	207	25.2 (20.8–30.3)	81	29.0 (24.3–34.2)	93	15.0 (11.4–19.3)	48	10.6 (7.7–14.5)	34
No	8.7 (6.5–11.4)	45	9.2 (7.0–12.1)	48	49.9 (45.6–54.2)	259	43.2 (39.0–47.5)	224	22.9 (19.5–26.8)	119	21.2 (17.9–24.9)	110	10.2 (7.9–13.1)	53
Don’t know	25 (12.4–44.0)	7	14.3 (5.5–32.5)	4	46.4 (29.2–64.6)	13	35.7 (20.4–54.7)	10	35.7 (20.4–54.7)	10	25.0 (12.4–44.0)	7	—	0

95%-CI: 95% confidence interval; GP: general practitioner.

## Data Availability

Due to ethical reasons data are not publicly available, however authors may be contacted for more information.

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
