# Peer review of "Perceived Access to Health Care Services and Relevance of Telemedicine during the COVID-19 Pandemic in Germany"

_ijerph, 2021, doi:10.3390/ijerph18147661_

Round 1

Reviewer 1 Report

The paper explains patients' perception of health services during Covid-19. I highlight some comments:

-Abstract. Line 24 line “In conclusion, for the majority of partecipants access to medical care was ensuring during COVID-19 pandemic” and also on line 405. This is in contrast to line 410 about “the reduced availability and utilization of check-up visits..”. Try please to explain better.

-The introduction is long, and should only introduce the content of the paper. Some appreciable concepts of the introduction should be put into discussion.

- E-health includes many forms of medical informatics: administrative ITC procedures, instrumentation software, aid to emerging countries especially through mobiles and telemedicine. The latter includes the integration hospital territory, doctor-patient teleconsultation and doctor-doctor teleconsultation for the benefit and patients. The paper talks about doctor-patients teleconsultation, but it should be emphasized that the telemedicine carried out on this project is based on telephone calls or video calls, without sending medical data or the possibility of follow-up.

-In the Materials and Methods I ask the authors to better explain the COVID-19 Snapshot Monitoring (COSMO) (from WHO Europe's Insights Unit and Health Emergencies Program) as so far it is not running in all localities and is not known by many.

-Results 3.4, line 263. “There were no relevant differencies in satisfaction between telephone and video consultation”. This is because teleconsultations were done only by voice without the exchange of medical images, correct? If so it needs to be underlined better in the last paragraph of 4.3. Perhaps in the discussion mention several different types of telemedicine.

- Discussion. Health conditions and psychotherapeutic treatments are cited in the paper. If possible, mention more also other medical or surgical treatments.

-Discussion, line 402. Not only hard to reach, ma elderly population have basic difficulties in interacting with a telephone or monitor with respect to young people.

- The authors made a great effort to collect a lot of useful data. It would be interesting in the future to compare the data obtained during the pandemic with similar groups of post-pandemic patients.

Author Response

Dear reviewer,

thank you very much for your remarks.

Please find attached the detailed answers to your comments.

Kind regards,

Lukas Reitzle on behalf of the authors

Reviewer 2 Report

Summary

The Authors examine perceived access to medical care during three different stages of lockdown over the course of the COVID-19 pandemic in 2020, i.e. during the “first lockdown”, in the “period of mild restrictions”, and during the “less strict second lock-down”. Additionally, the Authors present stratified results on the required treatment occasion as well as the use and satisfaction regarding telemedicine instead of physical visits for the stage of the second lockdown from a population-based perspective.

The study is based on data from the serial, cross-sectional COVID-19 Snapshot Monitoring (COSMO) online survey. The analysis is based on data from wave 6, wave 17, and wave 28. The sample for each wave is matching the general population in Germany in terms of age, gender, and residency in a German federal state. Participants for whom the initial question about currently required medical appointments was not applicable were excluded, resulting in a final sample size of 773 participants for wave 6, 941 participants for wave 17, and 868 participants for wave 28.

In waves 6, 17, and 28, participants were asked: "Are doctor visits and contacts necessary for you currently available?". In wave 28, participants were further asked: 1) “What is the reason for doctor visits or contacts that are currently necessary for you?” and 2) “Do you currently use telephone or telemedical contact options instead of visiting a doctor’s or psychotherapist’s practice?” If the latter question was answered with yes regarding the telephone or telemedical contact, participants were further asked: 3) "Are you satisfied with your telephone or telemedical consultation?”

Consistent to previous studies based on COSMO data, age was categorized into the following four categories: 18 to 29 years, 30 to 49 years, 50 to 64 years, and 65 to 74 years, and education was maintained in the assessed three categories: up to 9 years of education, at least 10 years of education (without general qualification for university entrance), and at least 10 years of education (with general qualification for university entrance). The presence of chronic conditions was assessed by asking: “Do you have a chronic disease?” In the Section “Discussion” results are briefly summarized and commented with respect to other studies surveyed.

Broad comments

While the research may result of interest to the readership of the journal, the manuscript needs structural revisions. Apart a certain number of typos, tables should be presented in a dedicated Appendix, and consequently comments should be improved and extended. Furthermore, the segmentation of the sample by age and education is made by considering heterogeneous class intervals, consequently statistics should be weighted accordingly. In some cases, asymptotic normality does not hold as the number of observations is lower than 100, consequently the method used to identify class intervals should be illustrated more in detail, as the assumption of normality in some cases may result misleading (consider, as an example, the last column of Table 3 and the second column of Table 4).

Many comments may be supported by hypothesis testing. Considering, as an example, the statement at p.5, "Regarding age groups, the proportion of participants reporting ensured access to medical appointments increased substantially with older age", a hypothesis test between two proportions may validate this assumption.

A pooled logistic regression is mentioned in the manuscript, but the regression output is not presented.

Minor comments

The quality of the manuscript is affected by a certain number of typos.  

In Figure 2, the interpretation of the scores from 2 to 6 should be explicited if there was a label associated.

Author Response

(The authors gave the same response as above.)

Round 2

Reviewer 2 Report

Broad comments

Notwithstanding some revisions in the new version of the manuscript, some areas of weakness in the analysis have not been removed. Specifically, Authors have not provided a satisfying answer to points 2 and 4 of their reply, and they have provided only a partial answer to points 1 and 5.

Point 2:  the segmentation of the sample by age and education is made by 
considering heterogeneous class intervals, consequently statistics should be weighted accordingly.

As an example, consider a variable that may assume values from 0 to 1. I choose to classifying observations in two classes, the first one including scores from 0 to 0.1 (say, class A) and the second one including scores from 0.1 to 1 (say, class B), then I observe 10 obs. in the first class and 90 obs. in the second class. Consequently, I compute the following relative frequencies: fA = 0.1, fB = 0.9. In the interpretation of results, I cannot say that observations are concentrated in class B, because by weighting the absolute frequencies by the inverse of the class width I obtain the same result, that is on average I find one observation for each 1% interval of the domain.

Accordingly, in lines 195-196 it is stated that one third of the sample was in the age  group 30-49. This is true, however the interval 30-49 is almost twice as the interval 18-29, 1.5 times the interval 50-64, etc. Then, by weighting the absolute frequencies by the inverse of the class width it emerges how the sample is almost uniformly distributed.

Point 4: I appreciate the change from "substantially" to "significantly", but "significantly" in statistics means that a test was performed and the two values resulted "significantly" different. Considering that those one reported are sample (not population) data, I am not sure that the difference between 84.4, 94.4 and 97.8 is significative. If Authors have the single records, they may compute the sample variances for each interval and test this hypotheses.  

Point 1. I appreciate that numerous typos has been corrected and I respect Authors'choice, however in my opinion it is rather difficult to follow the text in section 3.

Point 5. Thank you for having added in the supplementary materials the output of the logistic regression. However, it is not clear how the logistic regression adds value to the main narrative, as it is quoted only in line 226. Authors may try to connect better this interesting and more sophisticated analysis with the main text. 

Author Response

Dear reviewer,

thank you very much for your follow-up comments. Please find attached our response to your remarks.

Kind regards,

Lukas Reitzle on behalf of the authors
